# STOCHASTIC SUBGOAL REPRESENTATION FOR HIERARCHICAL REINFORCEMENT LEARNING

## ABSTRACT

Goal-conditioned hierarchical reinforcement learning (HRL) promises to make long-term decision-making feasible by reducing the effective planning horizon through a latent subgoal space for high-level policies. However, existing methods employ deterministic subgoal representations, which may hinder the stability and efficiency of hierarchical policy learning. This paper introduces a Gaussian process (GP)-based Bayesian approach to learn stochastic subgoal representations. Our method learns a posterior distribution over the latent subgoal space, utilizing GPs to account for the stochastic uncertainties in the learned representation, thus facilitating improved exploration. Moreover, our approach offers an adaptive memory that integrates long-range subgoal information from prior planning steps. This enhances representation in novel state regions and bolsters robustness against environmental stochasticity. In experiments, our approach surpasses state-of-the-art HRL methods in both deterministic and stochastic settings with dense and sparse external rewards. Additionally, we demonstrate that our approach allows transfer of low-level policies across tasks.

## 1 INTRODUCTION

Tackling complex problems with long-term credit assignment has been one of the major challenges for reinforcement learning (RL), and hierarchical deep reinforcement learning (HRL) has demonstrated remarkable capabilities in solving a wide range of temporally extended tasks with sparse rewards, by enabling control at multiple time scales via a hierarchical structure. Goal-conditioned HRL methods, in which the higher-level policies periodically set subgoals for lower-level policies and the lower level is intrinsically rewarded for reaching those subgoals, have long held the promise to be an effective paradigm in HRL (Dayan & Hinton, 1992; Schmidhuber & Wahnsiedler, 1993; Kulkarni et al., 2016; Vezhnevets et al., 2017; Nachum et al., 2018; Levy et al., 2019; Zhang et al., 2020; Li et al., 2021; 2022).

The subgoal representation function in goal-conditioned HRL maps the state space to a latent subgoal space. Learning an appropriate subgoal representation function is critical to the performance and stability of goal-conditioned HRL. Since the subgoal space corresponds to the high-level action space, the subgoal representation contributes to the stationarity of the high-level transition functions. Furthermore, the low-level reward function, *i.e.*, intrinsic rewards, is defined in latent subgoal space in goal-conditioned HRL, and low-level behaviors can be induced by dynamically changing subgoal space as well. As such, a proper abstract subgoal space contributes to the stationarity of hierarchical policy learning.

A wide variety of subgoal representations have been investigated, ranging from directly utilizing the state space (Levy et al., 2019) or hand-crafted space (Nachum et al., 2018), to end-to-end learning without explicit objectives (Vezhnevets et al., 2017) or deterministic representations learned by imposing local constraints (Li et al., 2021). However, none of the existing representations have explicitly modeled stochasticity and long-range historical latent subgoal information. Previous works, such as Li et al. (2021), have utilized deterministic subgoal representation functions, which lack the ability to incorporate exploration or stochastic uncertainty in the subgoal representation. This can limit the exploration of hierarchical policies, resulting in the agent getting stuck in a local optimum or converging to a suboptimal policy. Furthermore, when the agent encounters novel state regions, it may not have enough historical information to determine a suitable subgoal representation. In such cases, deterministic subgoal representation functions may underfit the learning objective, leading to an inability to accurately capture the underlying dynamics of the environment in those new state regions. This, in turn, can result in poor performance and impede the agent's ability to achieve its goals. Although the active exploration strategy proposed by Li et al. (2022) aims to mitigate these issues in Li et al. (2021), the inherent limitations of deterministic mapping and short-term smoothness

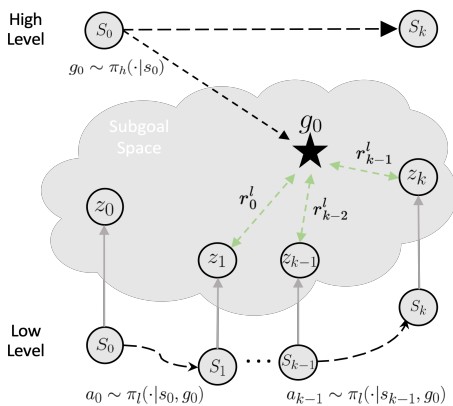

Figure 1: A schematic illustration of the hierarchical policy execution. One high-level step corresponds to k low-level steps. The negative Euclidean distance in the latent space provides intrinsic rewards for the low-level policy.

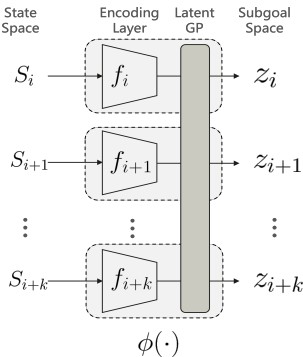

Figure 2: The representation function consists of an encoding layer and a latent GP layer. Taking as input the state $\mathbf{s}$, the encoding layer comprises a neural network to generate an intermediate latent space representation $\mathbf{f}$, which will be transformed by the GP layer to produce the final subgoal representation $\mathbf{z}$.

— arising from local constraints — still impede effective exploration and stationarity of training the hierarchical policy.

To address these limitations, we propose a novel GP based Bayesian approach for learning stochastic subgoal representations for goal-conditioned HRL (HSSR), which explicitly accounts for the uncertainties in the state space and allows for stable explorations with enhanced subgoal reachability. This is achieved by harnessing a nonparametric GP prior on latent subgoal space to learn a posterior distribution over the subgoal representation. Our approach explores the intrinsic structure in the state space through learnable kernels. It provides an adaptive memory that fuses subgoal information from previous planning steps, underpinning representations in novel state regions and offering resilience against environmental stochasticities. Leveraging the nature of Markov chains, we further present a lightweight subgoal representation formulation harnessing the state-space form GP, which efficiently fuses the long-range correlations in latent space from an arbitrary number of previous planning steps with a constant computational and memory complexity.

We benchmark our method on challenging continuous control tasks in both deterministic and stochastic settings with dense or sparse external rewards. Experimental results empirically demonstrate that our method is capable of generating stable stochastic subgoal representations which, on the one hand, contribute to the stationarity in both the high-level state transition and the low-level reward functions and, on the other hand, facilitates transferable low-level policies between tasks. The advantages of this first stochastic subgoal representation within HRL manifest as increased sample efficiency, heightened resilience against stochastic uncertainties, and a marked improvement in asymptotic performance when benchmarked against leading HRL methods.

## 2 PRELIMINARIES

The interaction between the agent and environment is generally modeled as a Markov Decision Process (MDP). Consider a goal-conditioned MDP which is represented by a tuple: MDP $=<\mathcal{S}, \mathcal{G}, \mathcal{A}, \mathcal{P}, \mathcal{R}, \gamma>$, where $\mathcal{S}$ is a state space, $\mathcal{G}$ is the subgoal set, $\mathcal{A}$ is an action set, $\mathcal{P} : \mathcal{S} \times \mathcal{A} \times \mathcal{S} \to [0, 1]$ is a state transition function, $\mathcal{R} : \mathcal{S} \times \mathcal{A} \to \mathbb{R}$ is a reward function, and $\gamma \in [0, 1)$ is a discount factor. We consider an HRL framework with two levels following Nachum et al. (2018) as illustrated in Fig. 1: the high-level policy $\pi_h(g|s)$ which operates at a coarser layer and generates a high-level action, *i.e.*, subgoal, and the low-level policy $\pi_l(a|s, g)$ which aims to achieve these subgoals. The high-level policy maximizes external reward by generating subgoals, *i.e.*, $g_i \sim \pi_h(\cdot|s_i) \in \mathcal{G}$, every $k$ timesteps when $i \equiv 0 \pmod{k}$. The low-level policy maximizes intrinsic reward associated with the subgoals by executing the primitive action.

## 3 METHOD

In this section, we present our Gaussian Process based stochastic subgoal representation. Firstly, we introduce a two-level goal-conditioned HRL framework with state-kernel GP prior. Then we present

GP latent-state batch estimation and training objective, which is followed by a lightweight online planning scheme.

## 3.1 Framework

We define the subgoal $g$ in the two-level HRL framework introduced by Nachum et al. (2018) in a low dimensional space abstracted by representation function $\phi(\mathbf{s}) : \mathbf{s} \mapsto \mathbb{R}^d$. Our method learns $\phi(\mathbf{s})$ simultaneously with the hierarchical policy. Specifically, we train the low-level policy $\pi_l(a|s, g)$ with an intrinsic reward function defined as the negative Euclidean distance in the latent subgoal space, *i.e.*, $r_l(s_i, a_i, s_{i+1}, g_i) = -||\phi(s_{i+1}) - g_i||_2$. The high-level policy is trained to maximize the extrinsic reward $r_i^h$ defined as $r_i^h = \sum_{t=i}^{i+k-1} r_t^{\text{env}}, i = 0, 1, 2, \cdots$, where $r_t^{\text{env}}$ is the reward from the environment. Our framework adopts the off-policy algorithm SAC (Haarnoja et al., 2018) for each level in the HRL structure, which generalizes the standard RL objective by augmenting it with an entropy term, *i.e.*, $\pi^* = \arg\max_{\pi} \sum_t \mathbb{E}_{(s_t, a_t) \sim \rho_\pi} [r(s_t, a_t) + \alpha \mathcal{H}(\pi(\cdot|s_t))]$. None-the-less, it is important to note that our method is agnostic to the specific HRL framework used. As illustrated in Fig. 2, the representation function $\phi(\mathbf{s})$ consists of an encoding layer and a latent GP layer. The encoding layer comprises a neural network to generate an intermediate latent space representation $\mathbf{f}$ by taking as input the state $\mathbf{s}$, which will be transformed by the GP layer to produce the final subgoal representation $\mathbf{z}$.

## 3.2 Stochastic Subgoal Representation

In order to specify a complete probabilistic model connecting state and subgoal spaces, a prior distribution for the latent subgoal $\mathbf{z}$ has to be defined. To this end, we impose GP priors to all $\mathbf{z}$ to model the stochastic uncertainties in subgoal space. Some of these uncertainties arise directly from environmental stochasticity, while others may be attributed to the unexplored regions of the state space. Specifically, we model the intermediate latent space representation $\mathbf{f}$ as a noise-corrupted version of the true latent subgoal space representation $\mathbf{z}$, and the inference can be stated as the following GP regression model:

$$\begin{aligned} \mathbf{z}_i &\sim \mathcal{GP}\left(0, \kappa\left(\mathbf{s}_i, \mathbf{s}_j\right)\right), \\ \mathbf{f}_i &= \mathbf{z}_i + \epsilon, \epsilon \sim \mathcal{N}(0, \sigma), \end{aligned} \tag{1}$$

where the noise variance $\sigma^2$ is a learnable parameter of the likelihood model, and $\kappa\left(\mathbf{s}_i, \mathbf{s}_j\right)$ is a positive-definite kernel function.

By modeling the uncertainties of subgoal space with GP priors, the mapping from state space to subgoal space is no longer a deterministic but a stochastic function to account for the full distribution of subgoal space. GP priors also define a probabilistic prior on the intermediate latent space which encodes for *a priori* knowledge that similar states should be mapped to more resembling latent subgoal representations than those mapped from distinct states. Such prior knowledge could be encoded by the kernel function, *i.e.*, $\kappa\left(\mathbf{s}_i, \mathbf{s}_j\right)$, defined over a distance in state space. Our insight is that the intrinsic structure in the state space could be exploited through learnable kernel function. We define the prior to be mean square continuous, once differentiable, and stationary in state space for the latent space processes (Williams & Rasmussen, 2006). Since the latent functions are intended to model the intrinsic structure of the state space, the latent space is expected to behave in a smooth and continuous fashion which is satisfied by Matérn kernel (Williams & Rasmussen, 2006),

$$\kappa\left(\mathbf{s}_i, \mathbf{s}_j\right) = \gamma^2 \left(1 + \frac{\sqrt{3}D\left(\mathbf{s}_i, \mathbf{s}_j\right)}{\ell}\right) \exp\left(-\frac{\sqrt{3}D\left(\mathbf{s}_i, \mathbf{s}_j\right)}{\ell}\right), \tag{2}$$

This kernel encodes the similarity between two states $\mathbf{s}_i$ and $\mathbf{s}_j$ in latent subgoal space subject to the distance function $D(\cdot)$ which is defined as $\ell^2$-norm. The learnable hyperparameters $\gamma^2$ and $\ell$ characterize the magnitude and length-scale of the processes respectively.

The inference problem in Eq. 1 can be solved for an unordered set of states, and the posterior mean and covariance are given by Williams & Rasmussen (2006):

$$\begin{aligned} \mathbb{E}[\mathbf{Z} \mid \mathbf{S}, \mathbf{F}] &= \mathbf{C}\left(\mathbf{C} + \sigma^2 \mathbf{I}\right)^{-1} \mathbf{F}, \\ \mathbb{V}[\mathbf{Z} \mid \mathbf{S}, \mathbf{F}] &= \text{diag}\left(\mathbf{C} - \mathbf{C}\left(\mathbf{C} + \sigma^2 \mathbf{I}\right)^{-1} \mathbf{C}\right), \end{aligned} \tag{3}$$

where $\mathbf{Z} = (\mathbf{z}_1 \, \mathbf{z}_2 \, \cdots \, \mathbf{z}_N)$ are the set of subgoal representations, $\mathbf{F} = (\mathbf{f}_1 \, \mathbf{f}_2 \, \cdots \, \mathbf{f}_N)$ are the set of intermediate latent representations from encoding layer, and $\mathbf{C}_{i,j} = \kappa\left(\mathbf{s_i}, \mathbf{s_j}\right)$ represents the covariance matrix. The true latent space representation, *i.e.*, the subgoal representation, $\mathbf{z}$ can be restored by taking the posterior mean of the GP.

### 3.3 LEARNING OBJECTIVE

In order to learn the hyperparameters of our stochastic subgoal representation, *i.e.*, $\sigma^2$, $\gamma^2$ and $\ell$, we propose a learning objective as follows:

$$\mathcal{L} = \frac{\Delta_{\mathbf{f}}^1}{\Delta_{\mathbf{f}}^k} \log(1 + \exp(\Delta_{\mathbf{z}}^1 - \Delta_{\mathbf{z}}^k)), \tag{4}$$

where $\Delta_{\mathbf{f}}^1 \propto ||\mathbf{f}_i - \mathbf{f}_{i+1}||$, $\Delta_{\mathbf{f}}^k \propto ||\mathbf{f}_i - \mathbf{f}_{i+k}||$, $\Delta_{\mathbf{z}}^1 \propto ||\mathbf{z}_i - \mathbf{z}_{i+1}||$ and $\Delta_{\mathbf{z}}^k \propto ||\mathbf{z}_i - \mathbf{z}_{i+k}||$. The logarithmic term in our proposed objective is designed to minimize the distance between low-level state transitions ($\Delta_{\mathbf{z}}^1$) in the latent subgoal space, while maximizing the distance for high-level state transitions ($\Delta_{\mathbf{z}}^k$). We employ the softplus function (Dugas et al., 2000) over the hinge loss for two main reasons. Firstly, it eliminates the need for a margin hyperparameter, thus simplifying the optimization process. Secondly, the softplus function provides continuous gradients, as opposed to the discontinuous gradients around margin planes seen in the hinge loss, facilitating finer adjustments within the subgoal space $\mathbf{Z}$. Furthermore, to enhance feature discrimination and the interaction between $\mathbf{F}$ and $\mathbf{Z}$, we use the ratio $\frac{\Delta_{\mathbf{f}}^1}{\Delta_{\mathbf{f}}^k}$ as a relative distance measure in $\mathbf{F}$ for the auxiliary loss.

This approach promotes closer intermediate latent representations for low-level state transitions with smaller ratios and greater separation for high-level transitions with larger ratios, focusing on the relative ratio rather than the absolute difference.

This objective is specifically designed for modeling the stochasticity of subgoal space (ratio term) while facilitating smooth and yet discriminative subgoal representation learning in GP latent space (logarithm term). Rather than learning a deterministic mapping from state space to subgoal space, our stochastic approach explicitly represents subgoals at a finite number of support points, *i.e.*, $\mathbf{S} = \{\mathbf{s}_i, \mathbf{s}_{i+1}, \mathbf{s}_{i+k}\}$, and let the GPs generalize to the entire space through the kernel function with learned hyperparameters.

### 3.4 EFFICIENT ONLINE SUBGOAL GENERATION

During learning, we proposed a batch solution for HRL with latent GP subgoals that considers all the interconnected states in the trajectory. However, the inference involves matrix inversion of the covariance matrix $\mathbf{C}$ which grows with the number of states in the trajectory. Consequently, the inference complexity scales cubically with the number of states in the trajectory. During online HRL planning, the subgoal representation corresponding to states in the low-level trajectory follows a natural ordering, and thus our model can be relaxed to a direct graph, *i.e.*, Markov chain. This formulation can be solved exactly without approximations by state-space form GP (Sarkka & Hartikainen, 2012; Sarkka et al., 2013) with a constant memory and computational complexity per state.

Specifically, the GP prior for latent subgoals can be transformed into a dynamical model for state-space GP inference, based on the hyperparameters $\gamma^2$, $\ell$ and $\sigma^2$ learned from training. The initial latent subgoal representation is estimated corresponding to Matérn covariance function, *i.e.*, $\mathbf{z}_0 \sim \mathcal{N}(\boldsymbol{\mu}_0, \boldsymbol{\Sigma}_0)$ where $\boldsymbol{\mu_0} = \mathbf{0}$ and $\boldsymbol{\Sigma_0} = \mathrm{diag}\left(\gamma^2, 3\gamma^2/\ell\right)$. As derived in Sarkka et al. (2013), an evolution operator which has the behavior of the Matérn kernel is defined:

$$\boldsymbol{\Psi}_i = \exp\left[\begin{pmatrix} 0 & 1 \\ -3/\ell^2 & -2\sqrt{3}/\ell \end{pmatrix} \Delta S_i\right], \tag{5}$$

where the state difference $\Delta S_i = D(\mathbf{s}_i, \mathbf{s}_{i-1})$ is the distance between consecutive states. Then the subgoal representation is predicted by $\mathbf{z}_i|\mathbf{f}_{1:i-1} \sim \mathcal{N}(\tilde{\boldsymbol{\mu}}_i, \tilde{\boldsymbol{\Sigma}}_i)$, where the mean and covariance are propagated as:

$$\tilde{\boldsymbol{\mu}}_i = \boldsymbol{\Psi}_i \boldsymbol{\mu}_{i-1}, \tag{6}$$

$$\tilde{\boldsymbol{\Sigma}}_i = \boldsymbol{\Psi}_i \boldsymbol{\Sigma}_{i-1} \boldsymbol{\Psi}_i^\top + \boldsymbol{\Omega}_i, \tag{7}$$

where $\boldsymbol{\Omega}_i = \boldsymbol{\Sigma}_0 - \boldsymbol{\Psi}_i \boldsymbol{\Sigma}_0 \boldsymbol{\Psi}_i^\top$. The posterior mean and covariance is conditioned on the current intermediate latent representation $\mathbf{f}_i$:

$$\boldsymbol{\mu}_i = \tilde{\boldsymbol{\mu}}_i + \mathbf{k}_i(f_i^\top - \mathbf{h}^\top \tilde{\boldsymbol{\mu}}_i), \tag{8}$$

$$\boldsymbol{\Sigma}_i = \tilde{\boldsymbol{\Sigma}}_i - \mathbf{k}_i \mathbf{h}^\top \tilde{\boldsymbol{\Sigma}}_i, \tag{9}$$

where $\mathbf{k}_i = \tilde{\boldsymbol{\Sigma}}_i \mathbf{h} / \left(\mathbf{h}^\top \tilde{\boldsymbol{\Sigma}}_i \mathbf{h} + \sigma^2\right)$ and the observation model $\mathbf{h} = (1\,0)^\top$. The derivation of the above recursive update for the posterior mean and covariance for a new state $\mathbf{s}_i$ can be found in the appendix. We note its resemblance to the Kalman Filter updates.

|  |  | HSSR | HESS | LESSON | HRAC | TD3 |
|---|---|---|---|---|---|---|
| Ant Maze | Dense | **0.90±0.04** | 0.86±0.01 | 0.81±0.04 | 0.76±0.06 | 0.00±0.00 |
|  | Sparse | **0.93±0.05** | 0.84±0.01 | 0.77±0.10 | 0.83±0.06 | 0.00±0.00 |
|  | Dense /w image | **0.83±0.06** | 0.78±0.05 | 0.73±0.05 | 0.00±0.00 | 0.00±0.00 |
|  | Sparse /w image | **0.79±0.07** | 0.67±0.12 | 0.71±0.05 | 0.00±0.00 | 0.00±0.00 |
| Ant Push | Dense | **0.93±0.01** | 0.80±0.04 | 0.71±0.02 | 0.01±0.00 | 0.00±0.00 |
|  | Sparse | **0.91±0.01** | 0.77±0.05 | 0.71±0.02 | 0.08±0.03 | 0.00±0.00 |
|  | Dense /w image | **0.84±0.05** | 0.70±0.03 | 0.24±0.01 | 0.01±0.01 | 0.00±0.00 |
|  | Sparse /w image | **0.87±0.03** | 0.73±0.06 | 0.67±0.03 | 0.00±0.00 | 0.00±0.00 |
| Ant Fall | Dense | **0.69±0.03** | 0.54±0.01 | 0.49±0.03 | 0.11±0.09 | 0.00±0.00 |
|  | Sparse | **0.79±0.01** | 0.29±0.05 | 0.54±0.02 | 0.24±0.07 | 0.00±0.00 |
|  | Dense /w image | **0.66±0.01** | 0.54±0.07 | 0.19±0.02 | 0.28±0.10 | 0.00±0.00 |
|  | Sparse /w image | **0.74±0.04** | 0.30±0.02 | 0.32±0.01 | 0.00±0.00 | 0.00±0.00 |
| Ant FourRooms | Dense | **0.93±0.02** | 0.80±0.01 | 0.76±0.03 | 0.65±0.03 | 0.00±0.00 |
|  | Sparse | **0.89±0.04** | 0.82±0.08 | 0.77±0.01 | 0.76±0.01 | 0.00±0.00 |
|  | Dense /w image | **0.61±0.02** | 0.42±0.06 | 0.34±0.04 | 0.00±0.00 | 0.00±0.00 |
|  | Sparse /w image | **0.57±0.03** | 0.42±0.07 | 0.21±0.01 | 0.00±0.00 | 0.00±0.00 |

Table 1: Final performance of the policy obtained after 10M steps of training, averaged over 10 randomly seeded trials with standard error. Comparisons are to **HESS** (Li et al., 2022) **LESSON** (Li et al., 2021), **HRAC** (Zhang et al., 2020), and "flat" RL TD3 (Fujimoto et al., 2018). We can observe the overall superior performance of our method in stochastic environments, with dense or sparse external rewards and with or without top-down image observations.

Note that the posterior latent subgoal representation $\mathbf{z}_i|\mathbf{f}_{1:i} \sim \mathcal{N}(\boldsymbol{\mu}_i, \boldsymbol{\Sigma}_i)$ is conditioned on all state till the current time step and thus is able to encode longer-term memory of high-level actions.

## 4 RELATED WORK

Goal-conditioned HRL (Vezhnevets et al., 2017; Nachum et al., 2018; Levy et al., 2019; Zhang et al., 2020; Wang et al., 2020; Li et al., 2021) where the high-level policy periodically generates subgoals to a low-level policy whilst the low-level policy learns how to efficiently reach these subgoals, has demonstrated great potentials in tackling temporally extended problems. A proper subgoal representation is crucial to goal-conditioned HRL since it defines the high-level action space and thus contributes to the stationarity of the high-level transition functions. Moreover, low-level behaviors can also be induced by dynamically changing subgoal space where the low-level reward function is defined. Hand-crafted space, *e.g.*, predefining a subset of the state space as the subgoal space, has been adopted (Nachum et al., 2018; Zhang et al., 2020). However, this approach requires domain knowledge and is limited to certain tasks. Using the whole state space has been investigated in Levy et al. (2019), which is unscalable to tasks with high-dimensional observations. Péré et al. (2018); Nasiriany et al. (2019); Nair & Finn (2020) have utilized variational autoencoder (VAE) (Kingma & Welling, 2014) to compress high-dimensional observations in an unsupervised way, which, however, is unable to encode the states of hierarchical temporal scales in HRL. Vezhnevets et al. (2017) and Dilokthanakul et al. (2019) have developed implicit subgoal representations by learning in end-to-end manner jointly with hierarchical policies. Sukhbaatar et al. (2018) developed a pre-training approach to learning subgoal representations via self-play. Nachum et al. (2019) introduced the NOR approach by learning subgoal representations bounding the sub-optimality of hierarchical policies. Li et al. (2021) developed a slowness objective for learning a deterministic subgoal representation function. Nevertheless, the existing methods have only proposed deterministic subgoal representations which may hinder effective explorations. Adopting the deterministic subgoal representation of Li et al. (2021), Li et al. (2022) developed an active exploration strategy to enhance the high-level exploration, by designing measures of novelty and potential for subgoals.

Gaussian processes, which encode flexible priors over functions, are a probabilistic machine learning paradigm (Williams & Rasmussen, 2006). GPs have been used in other latent variable modeling tasks in RL. In Engel et al. (2003), the use of GPs for solving the RL problem of value estimation was first introduced. Then Kuss & Rasmussen (2003) used GPs to model the system dynamics and the value function. Deisenroth et al. (2013) has also developed a GP based transition model of a model-based learning system which explicitly incorporates model uncertainty into long-term planning and controller learning to reduce the effects of model errors. Levine et al. (2011) proposed

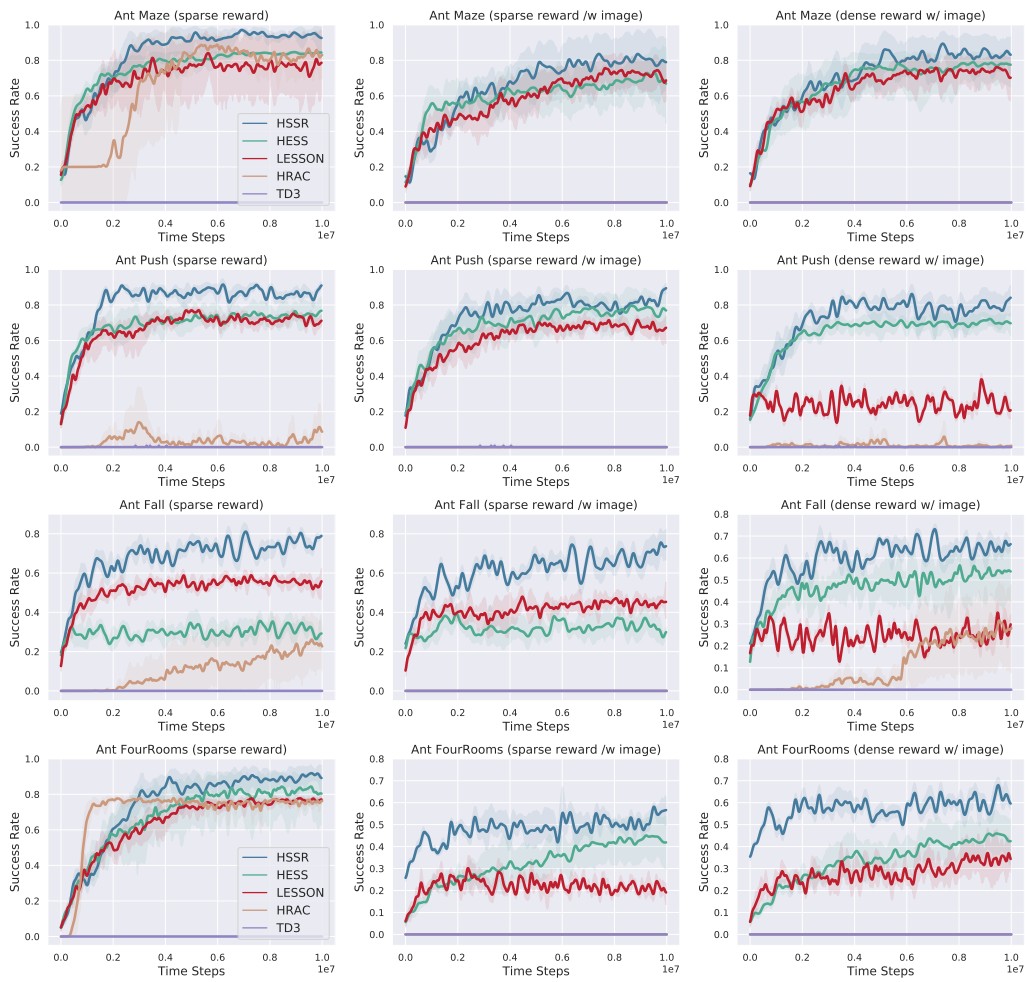

Figure 3: Learning curves of our method and baselines in **stochastic** environments, with dense or sparse external rewards and with or without top-down image observations. Each curve and its shaded region represent the average success rate and 95% confidence interval respectively, averaged over 10 independent trials. We find that our method performs well across all tasks. It is worth noting that our method learns both rapidly and stably; on complex navigation tasks, it normally requires only less than three million environment steps to achieve good performance.

an algorithm for inverse reinforcement learning that represents nonlinear reward functions with GPs, which was able to recover both a reward function and the hyperparameters of a kernel function that describes the structure of the reward.

## 5 EXPERIMENTS

We evaluate our method in challenging environments with dense and sparse external rewards which require a combination of locomotion and object manipulation to demonstrate the effectiveness and transferability of our learned stochastic subgoal representations. We compare our methods against standard RL and prior HRL methods. We also perform ablative studies to understand the importance of various components. Our experiments are designed to answer the following questions: (1) Can HSSR outperform state-of-the-art HRL methods in terms of stability, sample efficiency, and asymptotic performance? (2) Can the stochastic subgoal representation enhance robustness against environmental stochasticity? (3) Can HSSR induce reachable subgoals to mitigate the non-stationarity issue of off-policy training in HRL? (4) How do various design choices influence empirical evaluation?

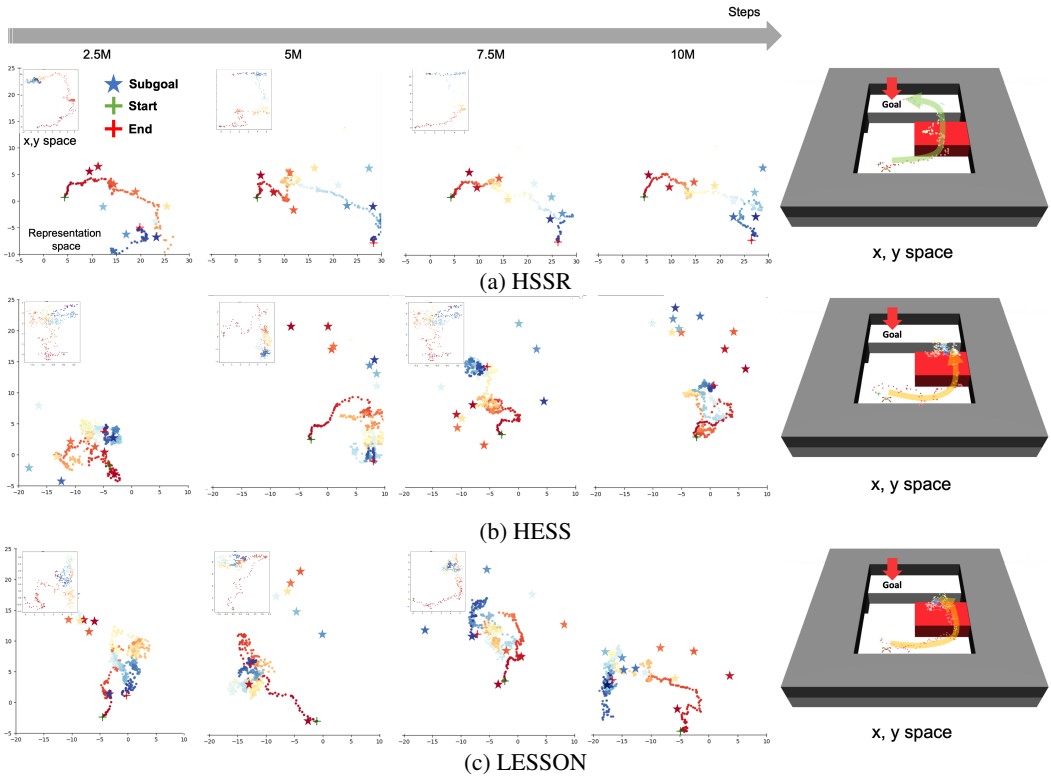

Figure 4: Subgoal representation learning process in the challenging stochastic Ant Fall task with sparse reward. Each figure contains the agent trajectory (from red to blue) in the representation space and the x, y space (top-left), as well as the subgoals (blue stars) in the latent space. HSSR consistently learns stable subgoal representations over training, compared to HESS and LESSON - from 5M steps until the end, there is no significant change in the latent space. The subgoals of HSSR in the latent space align with low-level trajectories, ensuring stable high-level transitions and low-level reward functions. In contrast, HESS and LESSON exhibit poor subgoal reachability. HESS uses the counts across dramatically changing representation as novelty measure which misleads the exploration and generates unreachable subgoals. Both LESSON and HESS struggle to learn stable deterministic representations imposing local constraint in the presence of environmental stochasticity. In HSSR, distances in the latent space correlate with global transition counts, ensuring a representative distance between the start and goal of the maze. This global perspective helps to mitigate the local optima observed in HESS and LESSON, which arise from the local constraints applied during the training of deterministic subgoal representations.

## 5.1 ENVIRONMENTS

We evaluate on standard MuJoCo (Todorov et al., 2012) tasks widely adopted in the HRL community which include Ant Maze, Ant Push, Ant Fall and Ant FourRooms, as well as four variants with low-resolution image observations. To evaluate the benefits of the proposed stochastic subgoal representation, we make these tasks more challenging in the following ways: (1) Environmental stochasticity: we enhance the robustness assessment of HSSR by introducing Gaussian noise with standard deviation $\sigma = 0.1$ to the $(x, y)$ position of the agent at each step, following the precedent set by recent works such as HIGL (?) and HRAC (Zhang et al., 2020). The results from deterministic environments are detailed in the appendix. (2) Definition of "success": we tighten the success criterion to being within an $\ell^2$ distance of 1.5 from the goal, compared to a distance of 5 in Nachum et al. (2018) and Zhang et al. (2020). (3) External rewards: unlike the exclusive use of dense external rewards in Nachum et al. (2018), Zhang et al. (2020), and Li et al. (2021), we also test settings with sparse external rewards, where a successful goal reach yields a reward of 1, and all other outcomes yield 0. (4) Random start/goal: Contrary to Li et al. (2022), where the agent has fixed start and target positions, our tasks feature randomly selected start and target locations during training. All methods undergo evaluation and comparison under the uniform task settings, ensuring a fair assessment [1].

---

[1]Further details, including environment specifics, source code, and parameter settings for experiment reproduction, are provided in the appendix.

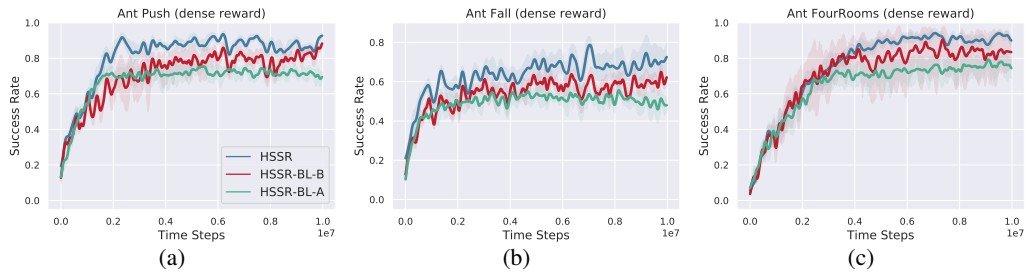

Figure 5: Ablation study comparing two baselines: HSSR-A omits our proposed learning objective and stochastic subgoal representation, while HSSR-B enhances HSSR-A by incorporating the proposed stochastic subgoal formulation and learning using an contrastive learning objective similar to that in Li et al. (2021).

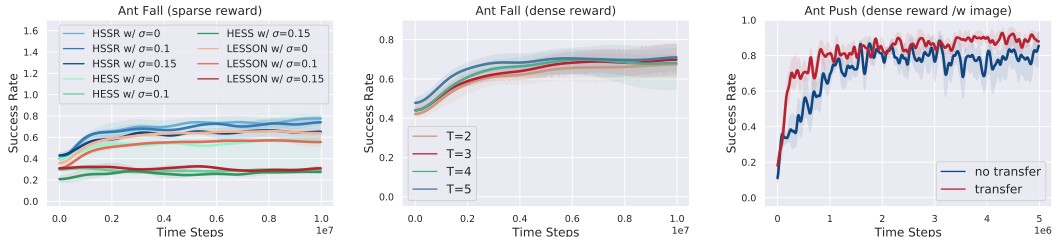

Figure 6: (Left) HSSR, HESS and LESSON on various levels of environmental stochasticities. (Middle) HSSR for various time window sizes of the state set from Eq. (3), used in batch estimation of model hyperparameters. All curves have been smoothed equally for visual clarity in the first two figures. (Right) Transfer learning for the task Ant Fall (Image) → Ant Push (Image). The transferred subgoal representation and low-level policy enable superior sample efficiency and enhanced asymptotic performance.

## 5.2 COMPARATIVE ANALYSIS

We conduct experiments comparing to the following state-of-the-art baseline methods[2]: (1) **LESSON** (Li et al., 2021): a HRL algorithm that learns the deterministic subgoal representation. (2) **HESS** (Li et al., 2022): a HRL algorithm which introduces an active exploration strategy to LESSON (Li et al., 2021). (3) **HRAC** (Zhang et al., 2020): a HRL algorithm which uses a pre-defined subgoal space. (4) **TD3** (Fujimoto et al., 2018): a flat RL algorithm to validate the need for hierarchical policies.

Table 1 shows the final performance of the trained policy. Our method significantly outperforms all compared baselines. Fig. 3 displays the learning curves of our method and baselines across all tasks, with the results for dense external rewards provided in the appendix. Our method out-performs all baselines in terms of stability, sample efficiency and asymptotic performance. The advantage of the stochastic subgoal representation is more pronounced in the challenging Ant Fall and Ant FourRooms tasks. Ant Fall requires both task and motion planning, while Ant FourRooms uses a larger scale maze. Thus both tasks demand learning subgoal representation for unexplored areas. In the tasks with image input, the benefit of stochastic subgoal representation of our method is more substantial, since learning the subgoal representation in a higher dimensional state space is more challenging and creates non-stationarities for deterministic subgoal representations in LESSON. The active exploration method introduced by HESS provides advantages in enhancing the generalization of deterministic subgoal representations in unexplored states (*e.g.*, Ant FourRooms with images) which is optimized for tasks with a fixed start and goal. However, its novelty measure, which combines counts in dynamically changing representation spaces, can potentially mislead exploration (as seen in Ant Fall), especially when the goal is random. The results show a clear advantage of learned subgoal representations (HSSR, LESSON and HESS) compared to pre-defined (HRAC) subgoal spaces. Finally, the flat RL algorithm TD3 does not learn in the complex environments used in the experiments which further validates the need for hierarchical policies.

Fig. 4 illustrates the state embeddings learned at various training stages for the challenging Ant Fall task with sparse external rewards. This allows for an intuitive comparison of subgoal representations acquired by HSSR, HESS, and LESSON. Notably, HSSR demonstrates stable evolution of subgoal representations throughout training, in contrast to HESS and LESSON. There are two in-depth

---

[2]We use the official implementations https://github.com/SiyuanLee/LESSON, https://github.com/SiyuanLee/HESS/, https://github.com/trzhang0116/HRAC and https://github.com/sfujim/TD3.

observations: (1) The subgoals in the latent spaces that HSSR learns are both reachable and largely align with the low-level trajectories. This suggests that a stable subgoal representation enhances the stationarity of the high-level transitions and the low-level reward functions, providing strong learning signal even at the early stage of training. On the other hand, HESS and LESSON exhibit unstable embeddings and frequently shift distant subgoals. (2) In HSSR, the Euclidean distances in the latent space roughly correspond to the total number of transitions. More precisely, considering the number of transitions necessary for the agent to navigate between them, the start and goal positions in the maze should be distinctly separated in the latent space. However, due to the local constraints applied to the deterministic subgoal representations in both HESS and LESSON, the start and goal locations remain closely associated in the latent space. Consequently, many intermediate embeddings become stuck in local optima because they lack the global constraint present in HSSR. We underscore that our stochastic representation learns the hyperparameters for the kernel function through finite number of support states, and then generalize to the entire space with a posterior distribution over the subgoal latent space.

## 5.3 Ablative Analysis

We conduct several ablation studies to analyze the design choices in our method. Initially, we compare our method, HSSR, with two baselines. HSSR-BL-A omits our proposed learning objective and stochastic subgoal representation. In contrast, HSSR-BL-B builds upon HSSR-BL-A by incorporating the proposed stochastic subgoal formulation and employing a contrastive learning objective akin to that used in Li et al. (2021). Fig. 5 shows the learning curves of various baselines. HSSR-BL-B exhibits much higher asymptotic performance than HSSR-BL-A but slightly lower performance than HSSR. This empirically demonstrates the effectiveness of our stochastic subgoal representation and learning objective respectively.

We evaluate the robustness of HSSR against various environmental stochasticities and compare its performance with the deterministic subgoal representation approach LESSON, as well as with HESS. As illustrated in Fig. 6 (Left), HSSR consistently outperforms both HESS and LESSON with increasing levels of Gaussian noise, specifically at $\sigma$ values from the set $(0, 0.1, 0.15)$. Notably, HSSR demonstrates significantly smaller degradation in performance and lower variance in outcomes as environmental stochasticity increases, compared to the observed results in HESS and LESSON.

We investigate the time window size of the set of states in Eq. (3) which are used to learn the model hyperparameters in batch estimation. As shown in Fig. 6 (Middle), increasing the time window size $T$ gives better performance at early training steps ($10^6 \sim 7 \times 10^6$) and eventually achieves similar performance as small time windows in larger training steps ($7 \times 10^6 \sim 10^7$). Our insight is that a larger time window gives more stable model hyperparameters with less training steps, which in turn induces sample-efficient stationarity of the policies due to stable subgoal representations. We report all other results based on time window $T = 3$ without loss of generality.

## 5.4 Transferability Analysis

The generality of our GP based subgoal representation learning framework underpins transferable subgoal space as well as the low-level policy between different tasks of the same agent. To empirically experiment its transferability, the subgoal representation network, *i.e.*, encoding layer and latent GP layer, and low-level policy network are initialized in a target task with the weights learned in a source task, with the rest of the network randomly initialized. Two pairs of source and target tasks, *i.e.*, Ant Fall → Ant Push and Ant Fall (Image) → Ant Push (Image), are experimented. The learning curves on those two tasks are shown in Fig. 6 (Right), and we can observe that with the transferred subgoal representation and low-level policy the agent is more sample efficient and able to achieve higher performance.

## 6 Conclusion

This paper proposes a novel Gaussian process based stochastic subgoal representation learning method for HRL. Rather than learning a deterministic mapping, as is done in existing approaches, our probabilistic model learns the posterior probability over the subgoal latent space. Thus our approach remains stable in unexplored state spaces leading to stationarity in both the high-level transitions and low-level reward function. We propose a new learning objective to jointly learn the model hyperparameters and hierarchical policies in an end-to-end framework. Experiments show that the proposed stochastic subgoal representation improves the sample efficiency, robustness against stochastic uncertainties and asymptotic performance. We also demonstrate that the learned stochastic subgoal representation enables transferable low-level policies between tasks.

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

## A  APPENDIX

### A.1  ALGORITHM

We provide Algorithm 1 to show the training procedure of HSSR. Some details of subgoal latent space formulation $z$ are omitted for brevity, which refers to Eq. (3). We provide the source code at https://anonymous.4open.science/r/HSSR-9EA6/.

---

**Algorithm 1:** HSSR

---

**Data:** High-level policy $\pi_{\theta_h}^h$, low-level policy $\pi_{\theta_l}^l$, encoding layer $f(\cdot)$, non-parametric latent GP layer with learnable hyperparameters ($\sigma^2$, $\gamma^2$ and $\ell$), GP update frequency $m$, higher-level action frequency $k$, number of training steps $N$, replay buffer $D$.

**for** $n = 1$ **to** $N$ **do**

    Apply policies $\pi_{\theta_l}^l$ and $\pi_{\theta_h}^h$, collect experience $(s_t, g_t, a_t, r_t, s_{t+1}, g_{t+1})$

    Compute intrinsic reward $r_l(s_t, a_t, s_{t+1}, g_t) = -||\phi(s_{t+1}) - g_t||_2$

    Update replay buffer $D$

    Update low-level policy $\pi_{\theta_l}^l$ and encoding layer $f(\cdot)$ with experience from replay buffer $D$ every timestep with Eq. (4)

    Update high-level policy $\pi_{\theta_h}^h$ with experience from replay buffer $D$ every $k$ timesteps

    Update latent layer hyperparameter with a batch of state transitions from replay buffer $D$ every $m$ timesteps with Eq. (4)

---

## A.2 ONLINE INFERENCE

The Gaussian process inference problem we formulated in the batch scheme can be rewritten in the form

$$
\begin{aligned}
\mathbf{z} &\sim \mathcal{GP}\left(0, \kappa\left(\mathbf{s}, \mathbf{s}'\right)\right) \\
\mathbf{f} &= \mathbf{H}\,\mathbf{z}(\mathbf{s}) + \epsilon, \epsilon \sim \mathcal{N}(0, \sigma),
\end{aligned}
\tag{10}
$$

where the linear operator $\mathbf{H}$ selects the training set inputs among the latent subgoal space values $\mathbf{H}\,\mathbf{z}(\mathbf{s}) = (\mathbf{z}(\mathbf{s_1}), ..., \mathbf{z}(\mathbf{s_N}))$. This problem can be seen as an infinite-dimensional version of the Bayesian linear regression problem:

$$
\begin{aligned}
\mathbf{z} &\sim \mathcal{N}\left(0, K\right) \\
\mathbf{f} &= \mathbf{H}\,\mathbf{z}(\mathbf{s}) + \epsilon
\end{aligned}
\tag{11}
$$

where $\mathbf{z}$ is a vector with Gaussian prior $\mathcal{N}\left(0, K\right)$ and $\mathbf{H}$ is constructed to select those elements of the vector $\mathbf{z}$ that can be actually observed (Sarkka & Hartikainen, 2012) .

This linear model can be extended such that the vector is allowed to change in time according to a linear stochastic differential equation (SDE) model and a new vector of measurements is obtained at times $t_i$ for $i = 1, ..., T$ (Särkkä & Solin, 2019):

$$
\begin{aligned}
\frac{\partial \mathbf{z}(\mathbf{t})}{\partial t} &= \mathbf{A}\,\mathbf{z}(t) + \mathbf{L}\mathbf{w}(t) \\
\mathbf{f}_i &= \mathbf{H}\,\mathbf{z}(t_i) + \epsilon_i,
\end{aligned}
\tag{12}
$$

where $i = 1, ...T$, $\mathbf{A}$, $\mathbf{L}$ and $\mathbf{H}$ are given matrices, $\epsilon_i$ is a vector of Gaussian measurements noises, and $\mathbf{w}(t)$ is a vector of white noise processes. The problem of estimating $\mathbf{z}(t)$ given the measurements can be solved using the classical Kalman filter and Rauch-Tung-Striebel (RTS) smoother. Assuming $\mathbf{z}(t_0) = \mathcal{N}(\boldsymbol{\mu}_0, \boldsymbol{\Sigma}_0)$, evolution operator $\boldsymbol{\Psi}_i$, and $\boldsymbol{\Omega}_i = \boldsymbol{\Sigma}_0 - \boldsymbol{\Psi}_i \boldsymbol{\Sigma}_0 \boldsymbol{\Psi}_i^{\top}$, the filtering solution is recursively given by the following Kalman filter (Sarkka et al., 2013):

- Prediction step:

$$
\begin{aligned}
\tilde{\boldsymbol{\mu}}_i &= \boldsymbol{\Psi}_{i-1}\boldsymbol{\mu}_{i-1}, \\
\tilde{\boldsymbol{\Sigma}}_i &= \boldsymbol{\Psi}_{i-1}\boldsymbol{\Sigma}_{i-1}\boldsymbol{\Psi}_{i-1}^{\top} + \boldsymbol{\Omega}_i
\end{aligned}
$$

- Update step:

$$
\begin{aligned}
\mathbf{v}_i &= \mathbf{y}_i - \mathbf{H}_i \tilde{\boldsymbol{\mu}}_i, \\
\mathbf{S}_i &= \mathbf{H}_i \tilde{\boldsymbol{\Sigma}}_i \mathbf{H}_i^{\top} + \sigma^2, \\
\mathbf{K}_i &= \tilde{\boldsymbol{\Sigma}}_i \mathbf{H}_i^{\top} \mathbf{S}_i^{-1} \\
\boldsymbol{\mu}_i &= \tilde{\boldsymbol{\mu}}_i + \mathbf{K}_i \mathbf{v}_i, \\
\boldsymbol{\Sigma}_i &= \tilde{\boldsymbol{\Sigma}}_i - \mathbf{K}_i \mathbf{S}_i \mathbf{K}_i^{\top}
\end{aligned}
$$

The subgoal representation during *online planning* can be formulated as spatio-temporal Gaussian process regression problem with models of the form

$$
\begin{aligned}
\mathbf{z}(\mathbf{s}, t) &\sim \mathcal{GP}\left(0, \kappa\left(\mathbf{s}, t; \mathbf{s}, t'\right)\right) \\
\mathbf{f}_i &= \mathbf{H}_i \mathbf{z}(\mathbf{s}, t_i) + \epsilon_i.
\end{aligned}
\tag{13}
$$

By representing the temporal correlation as a stochastic differential equation kind of model and the spatial dimension as an additional vector element index, it is equivalent to the infinite-dimensional state space model (Sarkka & Hartikainen, 2012) as counterpart of model Eq. 12:

$$
\begin{aligned}
\frac{\partial \mathbf{z}(\mathbf{s}, t)}{\partial t} &= \mathbf{A}\,\mathbf{z}(\mathbf{s}, t) + \mathbf{L}\mathbf{w}(\mathbf{s}, t) \\
\mathbf{f_i} &= \mathbf{H}_i\,\mathbf{z}(\mathbf{s}, t_i) + \epsilon_i,
\end{aligned}
\tag{14}
$$

where the latent state $\mathbf{z}(\mathbf{s}, t)$ at time $t$ consists of the whole function $\mathbf{s} \mapsto \mathbf{z}(\mathbf{s}, t)$, $\mathbf{A}$ is a $s \times s$ matrix of linear operators operating on $\mathbf{s}$, $\mathbf{L} \in \mathbb{R}^{s \times q}$, $\mathbf{H}_i \in \mathbb{R}^{d \times s}$ are given matrices, $\mathbf{f}_i \in \mathbb{R}^d$, $\epsilon_i \sim \mathcal{N}(0, \Sigma_i)$, and $\mathbf{w}(\mathbf{s}, t) \in \mathbb{R}^q$ is a Wiener process with a given diffusion matrix $\mathbf{Q}_c \in \mathbb{R}^{q \times q}$. This formulation is an infinite-dimensional Markovian type of model, where the problem of estimating $\mathbf{z}(\mathbf{s}, t)$ given the measurements can be similarly solved using the above Kalman filter resulting in the prediction and update steps in paper.

## A.3 ENVIRONMENTS

1. **Ant Maze** A '⊃'-shaped maze of size 12×12 for a quadruped-Ant to solve a navigation task. The ant needs to reach a goal position starting from a random position in a maze with dense rewards. It has a continuous state space including the current position and velocity, the current time step t, and the goal location. During training, a random position is generated as the goal for each episode, and at each time step the agent receives a dense or sparse reward according to its negative Euclidean distance from the goal position. The success is defined as being within an Euclidean distance of 1.5 from the goal. At evaluation stage, the goal position is set to (0, 8). Each episode ends at 500 time steps.

2. **Ant Push**: A challenging task that requires both task and motion planning. The agent needs to move to the left then move up and push the block to the right in order to reach the target.

3. **Ant Fall**: This task extends the navigation to three dimensions. The agent starts on a platform of height 4 with the target located across a chasm that it cannot cross by itself. The agent needs to push the block into the chasm and walk on top of it before navigating to the target.

4. **Ant FourRooms**: This Task requires the agent to navigate from one room to another to reach the exogenous goal. In this task, a larger ($18 \times 18$) maze structure is used.

5. **Variants**: Another Ant Maze of size $24 \times 24$ with the same definition of "success" is used (labeled 'Large'). A variant (labeled 'Image') with low-resolution image observations for each of the above task is adopted; the observation is formed by zeroing out the x, and y coordinates and appending a 5×5×3 top-down view of the environment, as described in Nachum et al. (2019); Li et al. (2021). Another variant with environmental stochasticity is also adopted - Gaussian noise with standard deviation $\sigma = 0.1$ to the $(x, y)$ position of the ant robot at every step is added.

## A.4 IMPLEMENTATION

### A.4.1 TRAINING AND EVALUATION PARAMETERS

- Learning rate of latent GP $1e - 5$
- Latent GP update frequency 100
- Batch GP scheme time window size 3
- Subgoal dimension of size 2
- Learning rate 0.0002 for actor/critic of both levels
- Interval of high-level actions $k = 50$
- Target network smoothing coefficient 0.005
- Reward scaling 0.1 for both levels
- Discount factor $\gamma = 0.99$ for both levels
- Learning rate for encoding layer 0.0001
- Hierarchical policies are evaluated every 25000 timesteps by averaging over 10 randomly seeded trials

### A.4.2 NETWORK ARCHITECTURES

We employ a two-layer hierarchical policy network similar to Levy et al. (2019); Li et al. (2021) which adopts SAC (Haarnoja et al., 2018) for each level in the HRL structure. Specifically, we adopt two networks each comprising three fully-connected layers (hidden layer dimension 256) with ReLU nonlinearities as the actor and critic networks of both low-level and high-level SAC networks. The output of the actor networks of both levels is activated using the tanh function and scaled according to the size of the environments. The encoding layer $f(\cdot)$ is parameterized by an MLP with one hidden layer of dimension 100 using ReLU activations. Adam optimizer is used for all networks.

### A.4.3 HARDWARE

All of the experiments were processed using a single GPU (Tesla V100) and 8 CPU cores (Intel Xeon Gold 6278C @ 2.60GHz) with 64 GB RAM.

|  | Ant Maze | Ant Maze (Large) | Ant Push | Ant Fall |
|---|---|---|---|---|
| HSSR | **0.96±0.00** | **0.93±0.03** | **0.90±0.01** | **0.74±0.02** |
| LESSON | 0.89±0.06 | 0.74±0.15 | 0.74±0.02 | 0.54±0.03 |
| HRAC | 0.90±0.03 | 0.83±0.03 | 0.01±0.00 | 0.45±0.08 |
| HIRO | 0.71±0.02 | 0.57±0.05 | 0.00±0.00 | 0.13±0.07 |
| ORACLE | 0.64±0.11 | 0.56±0.09 | 0.70±0.05 | 0.28±0.09 |
| TD3 | 0.00±0.00 | 0.00±0.00 | 0.01±0.00 | 0.00±0.00 |

Table 2: Final performance of the policy obtained after 5M steps of training in deterministic environments, averaged over 10 randomly seeded trials with standard error. Comparisons are to **LESSON** (Li et al., 2021), **HRAC** (Zhang et al., 2020), **HIRO** (Nachum et al., 2018), HRL with oracle subgoal space **Oracle**, and flat RL TD3 (Fujimoto et al., 2018). We can observe the overall superior performance of our method, which is consistent with the evaluation results in stochastic environments.

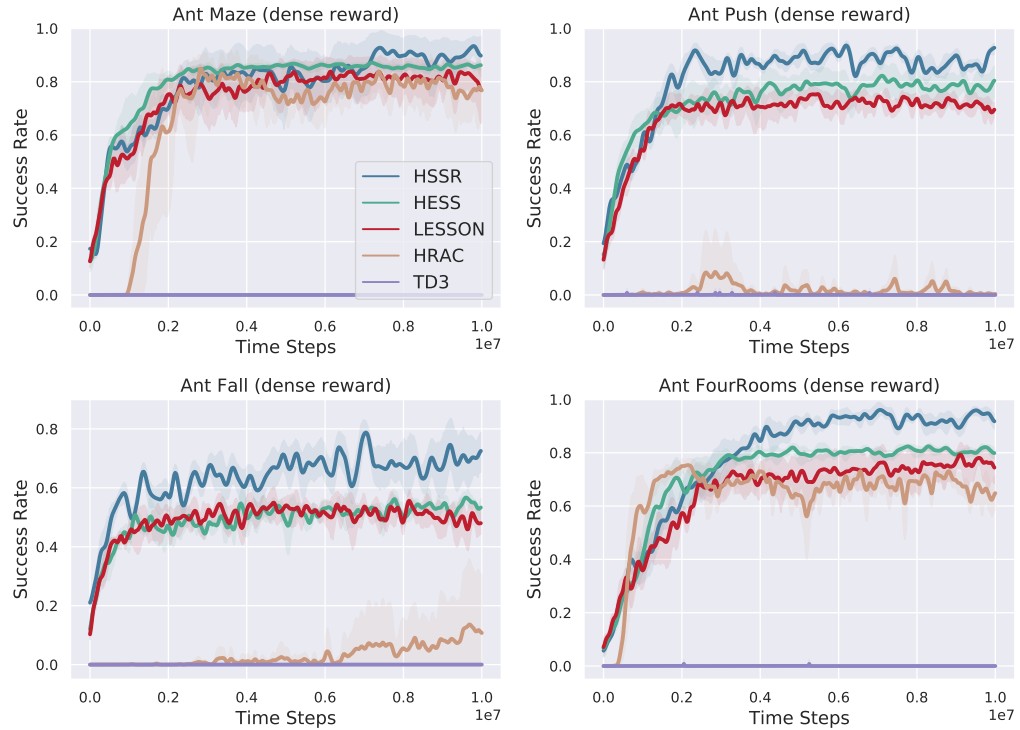

Figure 7: Learning curves of our method and baselines in **stochastic** environments with dense external rewards.

## A.5 Additional Experiments

We show the learning curves of our method and baselines in stochastic environments with dense external rewards in Fig. 7, and its quantitative evaluation results can be found in Table 1.

Additionally, we evaluate on deterministic Ant Maze, Ant Push and Ant Fall, as well as a 'large' Ant Maze of size $24 \times 24$, with dense external reward. These experiments are conducted in comparison to **LESSON** (Li et al., 2021), **HRAC** (Zhang et al., 2020) and **TD3** (Fujimoto et al., 2018), as well as the following two baseline methods:

1. **Oracle**: HRL with the oracle subgoal space, *i.e.*, $x$, $y$ coordinates of the agent, in navigation tasks.

2. **HIRO** (Nachum et al., 2018): an off-policy goal-conditioned HRL algorithm using a pre-defined subgoal space.

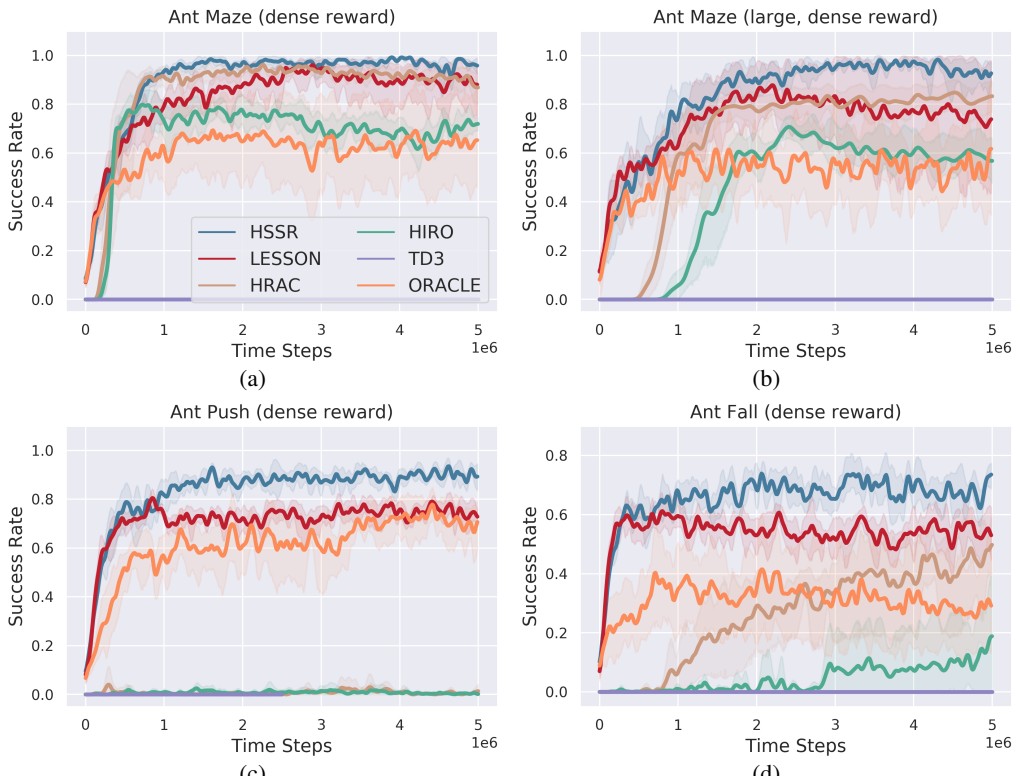

Figure 8: Learning curves of our method and baselines in **deterministic** environments with dense external rewards.

Note, all methods are evaluated and compared using the same settings of tasks. Table 2 shows the comparative results on deterministic environments, and Fig. 8 shows the learning curves of all baselines.

