# OpenReview forum: "Stochastic Subgoal Representation for Hierarchical Reinforcement Learning"
_ICLR.cc/2024/Conference — Submitted to ICLR 2024_

### Official Review · Reviewer_qYdp · 2023-10-21

**Soundness:** 3 good
**Presentation:** 3 good
**Contribution:** 3 good
**Rating:** 8
**Confidence:** 3

**Summary:**

This paper focuses on the problem of joint representation and policy learning in HRL. The proposed method, HAAR, introduces Gaussian Processes into representation learning, which effectively learns stochastic subgoal representations and outperforms current SOTA HRL algorithms.

**Strengths:**

1. The idea of learning stochastic subgoal representations is sensible and interesting.
2. Empirical results are strong and convincing.
3. Presentation is clear.

**Weaknesses:**

1. I am not very clear why the objective in Section 2.3 is designed like that. Why use a $log(1+exp())$ over $z$, while directly use a fraction over $z$? I think some additional discussion is needed.
2. I am not very familiar with GPs, and am interested in the computation complexity of HSSR. How much computation does it task compared to HESS?

**Questions:**

1. I am not very familiar with GPs, and am a bit confused about $f$ and $z$. From Fig 1 it seems that $f$ is an encoding of $s$, and is passed through GP to get $z$. However, Eq 1 seems to say that $f$ depends on $z$. Is it that HSSR first acquires $f$, and then denoises $f$ to get $z$?
2. How is HSSR compared to HESS if there is no environment randomness?

---

> ### Author Response · Authors · 2023-11-21
> **Response to Reviewer qYdp**
>
> **Q1: Learning objective**
>
> A1: Thank you for your insightful query regarding our learning objective's formulation. Our logarithmic term is based on the softplus function, as introduced in Dugas et al. [2001]. This choice offers two advantages over the more commonly used hinge loss function for contrastive learning:
> 1. **No Hyperparameter Requirement**: Unlike the hinge loss, the softplus function does not require setting a margin as a hyperparameter.
> 2. **Smooth Behavior**: The hinge loss exhibits discontinuity around margin planes, which can be problematic in GP regression. In contrast, the softplus function decays exponentially, avoiding the abrupt cut-off characteristic of the hinge function. This design ensures that even triplets meeting the margin criteria contribute to the loss, enabling continuous pushing/pulling samples as close or as far as possible. The goal is to regularize the absolute difference in $\mathbf{Z}$ space as the subgoal space.
> To further enhance feature discrimination and the interaction between $\mathbf{F}$ and $\mathbf{Z}$, we utilize the ratio as a relative distance measure in $\mathbf{F}$ as the auxiliary loss. This aims to encourage closer intermediate latent representations between low-level state transitions with smaller ratios, and to promote separation in the intermediate latent representations between high-level state transitions with larger ratios. It is the relative ratio that matters, as this term does not account for the absolute difference.
>
> **Reference:**
>
> Dugas C, Bengio Y, Bélisle F, Nadeau C, Garcia R. "Incorporating second-order functional knowledge for better option pricing." Advances in Neural Information Processing Systems. 2000;13.
>
> **Q2: relationship between f and z**
>
> A2: Thank you for your question regarding the role of $f$ and $z$ in the context of this paper. You are correct in your understanding that $f$ represents an intermediate encoding of the state space $s$, and $z$ is the latent subgoal embedding that we are ultimately interested in. The process flow involves mapping the original state space to an intermediate space $f$ through encoding, which is then used as an input to the GP to obtain $z$, i.e., the subgoal representation. The GP enables denoising the representation, taking into account the uncertainty and correlations in the input space. Eq. 1 provided defines a two-step process in a GP based model:
>
> 1. $z_i$, the subgoal representation of $s_i$, is governed by a Gaussian Process. This GP is characterized by a mean function, which is assumed to be zero, and a covariance function defined by the kernel $\kappa(s_i, s_j)$. The kernel function dictates the covariance or degree of similarity between any two points in the state space, thereby determining the relationship between the values of $z_i$ at different states $s_i$ and $s_j$.
>
> 2. $f_i$ is then defined as $z_i$ with an added Gaussian noise $\epsilon$, where $\epsilon \sim \mathcal{N}(0, \sigma)$. This introduces a stochastic element to the latent variable, creating the observed representation $f_i$ that encompasses the “stochasticity” not captured by the GP.
>
> The dependence of $f$ on $z$ indicates that $f$ is not just a direct observation from the state space but is a smoothed representation informed by the GP's interpretation of the state space. The addition of noise $\epsilon$ to form $f$ acknowledges that real-world observations may deviate from the GP's idealized $z$, reflecting the inherent uncertainty of real-world environments, capturing the unpredictability and randomness that are typical in practical scenarios. This addition not only improves the model's generalization capabilities to new or less frequent states but also facilitates exploration in the learning process, which is vital for effective policy development in HRL.
>
> **Q3: HSSR computational complexity, compared to HESS, without randomness**
>
> A3: In comparing the average inference time across 10,000 episodes using 5 different random seeds, HSSR exhibits an approximately 3% longer inference time than HESS.

---

> > ### Comment · Reviewer_qYdp · 2023-11-22
> > **Thanks for your response**
> >
> > Thank you for addressing my concerns. I would like to further clarify Q1. My question is, why do you use a fraction over $\Delta_f$, while using a softmax over  $\Delta_z$? Why not use the same form of loss for both variables? Is there any insight behind this, or is this just an empirical design?

---

> > ### Author Response · Authors · 2023-11-22
> > **Response to Reviewer qYdp**
> >
> > If the clarifications and revisions we've made address the concerns you highlighted, we would greatly appreciate your consideration for re-evaluating your initial rating. We look forward to any additional feedback you may have.

---

> > > ### Comment · Reviewer_qYdp · 2023-11-22
> > >
> > > I appreciate the author's efforts in addressing my concerns. Considering its contribution to the HRL community,  I am increasing my rating to 8.

---

> ### Author Response · Authors · 2023-11-22
> **Rationale Behind the Different Forms for $\mathbf{Z}$ and $\mathbf{F}$ Variables**
>
> Thank you for your further clarification on Q1. The choice of using a fraction for the variable $\mathbf{F}$ and a softplus for $ \mathbf{Z}$ in our loss function is based on specific insights related to the nature of these spaces:
> 1. **Subgoal Space $ \mathbf{Z}$ Regularization**: Our primary goal is to regularize the absolute distance (using softplus function) in the subgoal space $ \mathbf{Z}$. This is crucial since $\mathbf{Z}$ refers to the subgoal space, where maintaining and controlling the absolute distances is key to effective subgoal representation.
> 2. **Relative Distance in Intermediate Latent Space $\mathbf{F}$**: For $\mathbf{F}$, we use the ratio (fraction) as a relative distance measure to avoid two potential issues: (a) the need for an extra hyperparameter to balance absolute distances in both $\mathbf{Z}$ and $\mathbf{F}$ spaces, which would arise if the softplus function were used for both; and (b) the undue emphasis on the absolute distance in $\mathbf{F}$, a factor that is less crucial in our context. Our primary concern in $\mathbf{F}$ is the ratio between high-level and low-level transitions, effectively captured by a 'relaxed' relative measure rather than the absolute distances. This 'relaxed' approach provides more adaptable control over the transitions in $\mathbf{F}$.
>
> These design choices are driven by the distinct roles and requirements of the subgoal space and intermediate latent space, ensuring that each space is optimized in a way that best serves its purpose in the overall model. We hope the above explanation clarifies our choice regarding the two variables.

---

### Official Review · Reviewer_f9CU · 2023-10-31

**Soundness:** 3 good
**Presentation:** 3 good
**Contribution:** 3 good
**Rating:** 8
**Confidence:** 4

**Summary:**

This work proposes a novel method to produce subgoal representation in hierarchical RL. The authors show that the Gaussian Process can better form subgoal representations by learning a posterior distribution in the subgoal space. The authors theoretically analyze the learning process of the GP, and this analysis is accompanied by convincing experiments. The experiments and theoretical analysis show the effectiveness of the proposed approach.

**Strengths:**

1. This paper is well-structured. The authors first analyze the common limitations of existing HRL subgoal representation learning approaches and then provide solid theoretical and empirical evidence to show why and how the proposed method works, making this paper well understandable.

2. Experiments are well-described and highly reproducible. Experiments have good coverage. The selection of baselines and environments is reasonable and convincing.

**Weaknesses:**

1. Although this paper has a good choice of environments, the authors could also add some real-life applications (other than examples in MuJoCo) into their set of experiments to fully show the capability of the proposed method.

2. The explanation on how to learn the subgoal representation and perform GP inference is good. However, more theoretical analysis on why the idea of using GP in subgoal representation learning could work is appreciated.

**Questions:**

See weaknesses:

1. Can the authors provide more evidence (e.g. show some meaningful bounds) on why GP could help with subgoal representation learning?

2. Is it possible to test the proposed method in environments other than MuJoCo simulations? Some subgoal-sensitive tasks, such as NetHack or MineDojo could be interesting for experimenting.

---

> ### Author Response · Authors · 2023-11-21
> **Response to Reviewer f9CU**
>
> **Q1: Can the authors provide more evidence (e.g. show some meaningful bounds) on why GP could help with subgoal representation learning?**
>
> A1: Thank you for your question regarding the use of Gaussian Processes (GPs) in subgoal representation learning. We agree that providing more theoretical evidence, such as meaningful bounds, would strengthen our argument for using GPs in this context. We plan to include additional theoretical analysis in our future work to more rigorously demonstrate the advantages of GPs in modeling the latent subgoal space. Compared to existing approaches in learning subgoal representations for HRL, such as predefined subgoal spaces or deterministic representations learned via neural networks, our method introduces a significant advancement. We formulate a posterior distribution over the latent subgoal space, employing GPs to account for stochastic uncertainties in the learned representation. This approach allows for a more flexible and robust representation, adapting to the complex and uncertain environments typically encountered in HRL. We hope this response provides a clearer understanding of our approach and the future direction of our research in this area. Thank you again for your valuable feedback.
>
> **Q2: Is it possible to test the proposed method in environments other than MuJoCo simulations? Some subgoal-sensitive tasks, such as NetHack or MineDojo could be interesting for experimenting.**
>
> A2: Thank you for recommending that we extend our method's testing to environments beyond MuJoCo, such as NetHack and MineDojo. The advantage of using learnable subgoal representation is its adaptability to various settings without necessitating manual specification of subgoal environments. This flexibility allows our method to be effectively utilized in diverse environments, including NetHack and MineDojo.
>
>
> We chose MuJoCo for its established benchmarks in HRL research, following the lead of studies like HIRO, HRAC, and LESSON. While integrating new environments would greatly enhance our research, the limited timeframe of the rebuttal period makes this challenging. Comprehensive testing in additional environments requires significant time and resources for proper integration and analysis. We value your recommendation and plan to explore diverse environments in our future work to further validate and strengthen our method.

---

> ### Author Response · Authors · 2023-11-22
>
> If the clarifications and revisions we've made address the concerns you highlighted, we would greatly appreciate your consideration for re-evaluating your initial rating. We look forward to any additional feedback you may have.

---

> > ### Comment · Reviewer_f9CU · 2023-11-22
> >
> > Thank you for the answer. I acknowledge the clarifications by increasing my rating.

---

### Official Review · Reviewer_yjMq · 2023-11-06

**Soundness:** 3 good
**Presentation:** 3 good
**Contribution:** 3 good
**Rating:** 6
**Confidence:** 4

**Summary:**

This paper presents an approach named HSSR to learn representation of subgoals in Hierarchical RL (HRL) settings, when the environment and/or goal representation can be stochastic and of some degree of uncertainty. The main idea is to use Gaussian process (GP) where subgoal z is represented as a Gaussian Process sample; the GP kernel function (e.g. Matern kernel) is learned to capture the intrinsic structure of the (latent) state space. Learning of GP parameters can be done using Bayesian inference so as to satisfy temporal consistency in the low-level state transitions and maximize the distance in the high-level state transitions. The resulting algorithm, HSSR, can learn latent subgoal representations in stochastic (and deterministic) environments efficiently.

**Strengths:**

1. This paper studies an interesting and practically meaningful setting where environments can be stochastic and contains uncertainty, which were not studied extensively in the previous works. This brings novelty and interesting contributions to the field of representation learning for RL.


2. The choice of Matern kernel for Gaussian Process sounds good and looks reasonable; it allows a nice, learnable control of smoothness and continuity. The formulation makes sense and learning of Gaussian kernel hyperparameters based on the learning objective is simple, technically sound, and well-motivated.


3. Overall, experimental results seem good and promising. The downstream task learning performance with the representation learned can outperform existing baselines. I find the choice of baselines and evaluation settings reasonable.

**Weaknesses:**

- Although the motivation of dealing with stochasticity and uncertainty is interesting, I find the experimental setup of stochastic environments a bit artificial: the observations are just added Gaussian noises with \sigma=0.1 to the (x, y) position only. This seems to be an unrealistic setting that might be too advantageous for the method.

- First of all, it seems that the experiment settings can only deal with local, independent Gaussian noises rather than the inherent uncertainty that is a result of lack of exploration (i.e. unexplored state space).

- Second, why not perturbing the entire state space? I'm also not sure how much stochasticity \sigma=0.1 adds -- Figure 6(Left) suggest that LESSON with \sigma=0.1 doesn't degrade as much compared to the fully deterministic case -- \sigma=0.0, whereas 0.15 can significantly hurt LESSON; so I doubt that sigma=0.1 is a noise that is high enough that would make existing baselines. I think this work should show more clear evidence that HSSR can deal with stochastic noise compared to the baselines (than what is shown in Figure 6).


- Minor: Plots (especially Figure 6) are difficult to read, the plot should introduce more smoothing and post-processing. Regarding Figure 6(Left), it's very difficult to examine the effect of the stochasticity (different \sigma values).


- Minor nitpicking: The introduction section deserves a separate, numbered section. "Preliminaries" should start as a section 2, and 3. Method, etc.
- Minor nitpicking: The Matern kernel should have an equation number, say (2). Many equations throughout the paper (especially in Section 2.4) also lack the equation number.


- Section 2.3 and Equation (3) could benefit from more clarity: e.g. writing more explicitly what each of the term is trying to do. For instance, "minimizes the distance between low-level state transitions in the latent subgoal space" could be accompanied more directly with ||z_i - z_{i+1}||, etc.

**Questions:**

- The learning objective given in Equation (3) seems a bit arbitrary. Can the authors shed more light on why it has to be such a particular form? For instance, why are the \Delta_f terms given in the fractions? Why not simply have the form like \Delta_f^1  - \Delta_f^k + \Delta_z^1 - \Delta_z^k with some weight coefficients?


- Figure 6(Left) should also include HESS which is the strongest baseline other than LESSON. Why was HESS missing here?

---

> ### Author Response · Authors · 2023-11-21
> **Response to Reviewer yjMq, part 1**
>
> **Q1: setup of stochastic environments being a bit artificial; why not perturbing the entire state space?**
>
> A1: Thank you for your valuable feedback regarding our experimental setup, particularly our approach to simulating environmental stochasticity. Our study follows the precedent set by recent works such as **HIGL (Kim et al., 2021)** and **HRAC (Zhang et al., 2020)**. Specifically, HIGL added Gaussian noise at $\sigma=0.05$ to the $(x, y)$ position to model stochastic environments. Similarly, HRAC experimented with adding Gaussian noise levels of $\sigma=0.01, 0.05,$ and $0.1$ to the $(x, y)$ position, describing $\sigma=0.1$ as the "most noisy scenario."
>
> In our experiments, we consistently applied Gaussian noise at $\sigma=0.1$ across all tests, adhering to these established benchmarks. We further extended our investigation to include a $\sigma=0.15$ setting, as shown in Fig. 6 (Left). This decision aimed to thoroughly test our method's performance in highly stochastic settings, which are arguably more demanding than those used in previous studies.
>
> Your suggestion to perturb the entire state space in future work is insightful, yet it requires careful consideration. While we recognize that adding Gaussian noise to the $(x, y)$ positions as a proxy for environmental stochasticity is a simplification, these coordinates are the primary dimensions that are directly influenced by such stochasticity. Other dimensions, like torso contact forces, joint angles, and velocity, pose challenges in quantifying their respective impacts from the same level of stochasticity. Emphasizing the need for extensive experimentation with hyperparameters and design choices, such an investigation goes beyond the scope of this paper. It would require a dedicated research effort to adequately address these complexities. We hope this clarification provides a better understanding of our experiment settings and their context within the broader scope of HRL research. We remain open to further discussion and are grateful for the opportunity to enhance our work based on your insights.
>
> **Q2: the experiment settings can only deal with local, independent Gaussian noises rather than the inherent uncertainty that is a result of lack of exploration (i.e. unexplored state space).**
>
> A2: Our approach, which learns a posterior distribution over the subgoal representation, is specifically designed to address the limitations you've highlighted. Traditional deterministic subgoal representation methods often struggle in novel state regions due to insufficient historical information to form accurate subgoal representations. This can lead to an underfitting of the learning objective and an inability to accurately capture the underlying dynamics in these new areas of the state space. In contrast, our method explores the intrinsic structure in the state space through learnable kernels, and forms a posterior distribution over the subgoal representation. This approach is particularly effective in underpinning representations in novel state regions, offering a significant degree of resilience against environmental stochasticity, including those inherent uncertainties induced by unexplored spaces.
>
>
> To empirically validate our approach, we conducted comprehensive experiments in deterministic environments, as detailed in Fig. 8 (Appendix Section) of our paper. The results clearly demonstrate that our stochastic subgoal representation method offers performance improvements in handling the inherent uncertainty induced by unexplored state spaces, compared to the deterministic subgoal representation method LESSON. We hope this explanation clarifies how our method effectively deals with both types of uncertainties – the local, independent Gaussian noises and the inherent uncertainty from unexplored state regions.

---

> ### Author Response · Authors · 2023-11-21
> **Response to Reviewer yjMq, part 2**
>
> **Q3: Figure 6(Left) suggest that LESSON with \sigma=0.1 doesn't degrade as much compared to the fully deterministic case -- \sigma=0.0, whereas 0.15 can significantly hurt LESSON; so I doubt that sigma=0.1 is a noise that is high enough that would make existing baselines. Figure 6(Left) should also include HESS which is the strongest baseline other than LESSON. HESS is missing in Figure 6(Left)**
>
> A3: Indeed, LESSON with $\sigma=0.1$ doesn't degrade as much compared to the $\sigma=0.0$ case in this dense reward setting. However, as illustrated in Fig. 6 (Left) of the revision, such degradation is more pronounced in the sparse external reward setting, which is the primary setting also investigated in Fig. 5. Due to the lack of dense learning signals and the increasing presence of environmental stochasticity, HESS and LESSON start to show deteriorating degradations compared with HSSR.
>
> Our initial comparison didn’t include HESS as it is an extension of LESSON, incorporating an active exploration strategy on top of the deterministic subgoal representation method proposed by LESSON. Our objective was to isolate the impact of the stochastic vs. deterministic nature of subgoal representation methods in the face of environmental uncertainties, without the additional complexity introduced by active exploration strategies like those in HESS. In the revised version, we have now incorporated HESS into Figure 6 (Left) to ensure a more comprehensive comparison.
>
> **Q4: learning objective formulation**
>
> A4: Our logarithmic term is based on the softplus function, as introduced in Dugas et al. [2001]. This choice offers two advantages over the more commonly used hinge loss function for contrastive learning:
> 1. **No Hyperparameter Requirement**: Unlike the hinge loss, the softplus function does not require setting a margin as a hyperparameter.
> 2. **Smooth Behavior**: The hinge loss exhibits discontinuity around margin planes, which can be problematic in GP regression. In contrast, the softplus function decays exponentially, avoiding the abrupt cut-off characteristic of the hinge function. This design ensures that even triplets meeting the margin criteria contribute to the loss, enabling continuous pushing/pulling samples as close or as far as possible. The goal is to regularize the absolute distance in $\mathbf{Z}$ space as the subgoal space.
>
> For $\mathbf{F}$, we use the ratio (fraction) as a relative distance measure to avoid two potential issues: (a) the need for an extra hyperparameter to balance absolute distances in both $\mathbf{Z}$ and $\mathbf{F}$ spaces, which would arise if the softplus function were used for both; and (b) the undue emphasis on the absolute distance in $\mathbf{F}$, a factor that is less crucial in our context. Our primary concern in $\mathbf{F}$ is the ratio between high-level and low-level transitions, effectively captured by a 'relaxed' relative measure rather than the absolute distances. This 'relaxed' approach provides more adaptable control over the transitions in $\mathbf{F}$.
>
> These design choices are driven by the distinct roles and requirements of the subgoal space and intermediate latent space, ensuring that each space is optimized in a way that best serves its purpose in the overall model. We hope the above explanation clarifies our choice regarding the two variables.
>
> A straightforward formulation like $\Delta_\mathbf{f}^1 - \Delta_\mathbf{f}^k + \Delta_\mathbf{z}^1 - \Delta_\mathbf{z}^k$ with weight coefficients would linearly combine absolute distances in two spaces. This approach might not capture the intricate distances in either space effectively and could lead to excessive hyperparameter tuning and unstable training.
>
> **Reference:**
>
> Dugas C, Bengio Y, Bélisle F, Nadeau C, Garcia R. "Incorporating second-order functional knowledge for better option pricing." Advances in Neural Information Processing Systems. 2000;13.

---

> ### Author Response · Authors · 2023-11-22
> **Response to Reviewer yjMq**
>
> Following your constructive comments, we have provided detailed responses and made targeted revisions to our manuscript. If our clarifications have resolved the issues you raised, we would be thankful for a re-evaluation of your initial rating. Your further feedback is always welcome.

---

> > ### Comment · Reviewer_yjMq · 2023-11-22
> > **Thanks for posting author response**
> >
> > Dear authors, thank you very much for posting the author rebuttal and updating the draft with lots of improvements and clarifications.
> >
> > Q1:
> >
> > > Our study follows the precedent set by recent works such as HIGL (Kim et al., 2021) and HRAC (Zhang et al., 2020)
> >
> > Thanks for the discussion about perturbation of the entire state space and justification on the environment setup regarding stochasticity.
> >
> > If this is a common evaluation protocol agreed by the community and following some existing works, this **must be** cited and discussed in a self-contained manner. The draft has no reference to HIGL. I can see that HRAC (Zhang et al. 2020) uses a similar setup σ = 0.01, σ = 0.05, to make the environment *locally* stochastic. I would also argue that this is also an artificial and contrived setup. However, the focus of HRAC is not mainly on the stochastic environments unlike your work; the degree and quality of stochasticity would be much more important and crucial than previous works, so I think a more careful setup and choice of stochastic environments would be required.
> >
> > Although this arguably might not be a significant reason to reject the paper, it must be pointed out that this is an unrealistic simplification or some kind of assumptions being made. In real-world robotics tasks, dimensions pertaining to "non-goals" can also be noisy and stochastic, and the method may not work properly (as you said it requires more extensive study) in those settings where .
> >
> >
> > Q3. Figure 6 lacks HESS
> >
> > I can see that Figure 6(a) now includes HESS, thanks for addressing these comments.
> >
> > minor: Figure 6(a)(b) now have added smoothing as per my suggestion, but this doesn't look consistent with other plots including Figure 6(c). In the camera ready version, all the plot should have **the same** smoothing configuration for a better consistency. Also the colors (there are nine lines!) in Figure 6(a) are not very distinctive. I strongly suggest the authors to break this plot down in to several plots to more clearly deliver the message.
> >
> >
> > Q4. Loss function design, Eq.(4)
> >
> > Thanks for the further discussion and the insight on the softplus objective design and adding the discussion in the revised draft. This is very helpful.
> >
> > Ideally, it would be even better if there are some ablation studies to confirm the straightforward formulation or contrastive objectives would fail or work suboptimally.
> >
> >
> > Overall, the paper has made several improvements and I still believe the merits, idea, contributions are overweighing despite some concerns that are still remaining, about the limitation and scope of the stochastic environment. I'd like to retain my rating (6). Thank you.

---

> ### Author Response · Authors · 2023-11-23
>
> **Q1: Environment settings**
>
> A1: Thank you again for your valuable feedback. We have updated our manuscript to include references to both HIGL and HRAC in the context of stochastic environment settings. Given that our work is one of the first to concentrate on stochastic subgoal representation learning in HRL, we find the lack of established benchmarks in this specific area. Hence, we opted to enhance the noise configuration adopted in existing HRL works to rigorously test our method's resilience and the performance deterioration in baseline methods under increased stochasticity.
>
> Moreover, we emphasize that environmental stochasticity is only one of several challenges addressed in our evaluations. Other aspects, such as sparse external rewards and the use of image inputs, also contribute to the complexity of learning subgoal representations. Our method demonstrates its merits and advantages in these varied and challenging settings, making it arguably one of the most comprehensive among existing HRL works.
>
> We agree that real-world robotic tasks often encounter noise in multiple dimensions, presenting a broader challenge. Our reference to “more extensive study” pertains to future research exploring the quantification of noise level for each dimension of the state space, which falls outside the current scope of our work.
>
> **Q2: Figure plotting**
>
> We are grateful for your suggestion to enhance the quality of our figures. In the camera ready version, we will ensure consistent smoothing, use distinctive colors, and provide clear breakdowns in our figures for improved visual clarity and consistency.
>
> **Q3: Ablation study on loss function**
>
> A3: Regarding the ablation study on the loss function, Section 5.3 introduces HSSR-BL-B as a baseline model utilizing a contrastive learning objective, similar to that in LESSON. In Fig. 5, we present a comparison between HSSR-BL-B and HSSR, showcasing the advantages of our proposed learning objective that integrates the softplus function.

---

### Official Review · Reviewer_WYwK · 2023-11-09

**Soundness:** 1 poor
**Presentation:** 1 poor
**Contribution:** 2 fair
**Rating:** 1
**Confidence:** 4

**Summary:**

This paper proposes using the Gaussian Process as a non-parametric model for estimating the latent subgoal in Hierarchical RL. Specifically, the "intermediate subgoal representation" $f$ is assumed to be generated by injecting Gaussian noises to the "latent subgoals" $z$, which follows a Gaussian Process prior. This subgoal representation is then assumed to be estimated from state $s$, from which a posterior of the latent subgoal $z$ should be inferred. The authors propose an objective to learn the hyperparameter for the Gaussian process. However, they didn't include the training objective of RL. They reported results on several Mujoco tasks and claimed that they achieved better performance than existing methods.

**Strengths:**

This paper aims to tackle the stochasticity in the subgoal policy, which according to the authors' literature review, is an interesting problem.

The results look promising with sufficient ablation and visualization.

**Weaknesses:**

This paper is hard to understand without introducing their algorithm. The review felt confused about the poorly notations such as $\phi(s)$, $Z=[z_1 z_2 ... z_N]$, $\Delta_f$, $\Delta_z$, etc. The disturbing technical communication is below the standard of ICLR.

**Questions:**

What is the proposed complete algorithm?

Is the number of subgoals fixed? What is this number?

What is the relationship between $g$ and $z$?

How to interpret Fig.4 without marking which goal is associated with which segment of the trajectories?

---

> ### Author Response · Authors · 2023-11-21
> **Response to Reviewer WYwK**
>
> **Q1: This paper is hard to understand without introducing their algorithm. The review felt confused about the poorly notations such as \phi(s), etc. The disturbing technical communication is below the standard of ICLR.**
>
> A1: While all other three reviewers appreciated the technical clarity of the paper **(presentation: good, good, good)**, we acknowledge the importance of clarity in presenting and the reviewer’s suggestions. We have improved the paper and answer below each question in detail. Hope this satisfies the reviewer. We are more than willing to improve the paper writing and presentation further if needed.
>
> **Q2: What is the proposed complete algorithm?**
>
> A2: Due to the limited space in the main manuscript the detailed algorithm is provided in Appendix Section A.1. In the main paper, in Section 3.1, we introduce our two-level HRL framework, with Soft Actor Critic (SAC) being utilized at each HRL level. We have included the SAC RL training objective in the updated manuscript (Sec. 3.1) to improve clarity. We hope this clarifies your concern.
>
> **Q3: Poor notations**
>
> A3: In light of your comments, we wish to emphasize that our notation is broadly in line with established conventions in HRL and GP literatures. To clarify:
> - The notation $\phi(s)$ aligns with established conventions in HRL literature, as used in HESS [Li et al., 2022] (3rd line, last paragraph on page 2) and LESSON [Li et al., 2021] (4th line, 4th paragraph on page 2).
> - $\mathbf{Z} = \left( \mathbf{z}_1, \mathbf{z}_2, \cdots, \mathbf{z}_N \right)$ is similarly defined in [1] (3rd line, first paragraph on page 5).
> - $\Delta_\mathbf{f}$ and $\Delta_\mathbf{z}$ are shorthand notations representing distances, used for brevity. These forms are consistent with conventions found in algebra literature, such as those detailed in Section 3.2 of [2].
>
> We hope this clarification helps, and we are open to further suggestions to improve our manuscript.
>
> **References:**
> - [1] Lee, J., Bahri, Y., Novak, R., Schoenholz, S.S., Pennington, J. and Sohl-Dickstein, J., 2017. Deep neural networks as Gaussian processes. ICLR 2018.
> - [2] Berg, T., 2020. Intermediate Algebra. Kwantlen Polytechnic University.
>
> **Q4: Is the number of subgoals fixed? What is this number?**
>
> A4: Thank you for inquiring about the number of subgoals. In our experiments, the high-level policy is designed to generate a subgoal every $k$ timesteps, with $k$ set to 50 for all MuJoCo tasks, as detailed in A.4.1 in the Appendix section.
>
> **Q5: What is the relationship between g and z?**
>
> A5: Your question brings up an important aspect regarding the relationship between $g$ and $z$. In our framework, the high-level policy samples a subgoal $g_t$ from the latent subgoal space abstracted by the representation function $\phi(s)$. The low-level controller then strives to achieve these subgoals, with $z_t = \phi(s_t)$ serving as the mapping from the state space to the subgoal space. Essentially, $g$ represents the action taken by the high-level policy within the subgoal space, while $z$ is the corresponding subgoal space representation of the state $s$, as illustrated in Fig. 1.
>
>
> **Q6: How to interpret Fig.4 without marking which goal is associated with which segment of the trajectories?**
>
> A6: We are grateful for your constructive feedback regarding Fig. 4. Acknowledging the confusion caused by our initial presentation, we have revised the figure to enhance clarity. The subgoals are now color-coded to match their corresponding states, facilitating a more intuitive understanding of their associations with specific trajectory segments.

---

> ### Author Response · Authors · 2023-11-22
> **Response to Reviewer WYwK**
>
> If we have adequately addressed your concerns, we would greatly appreciate a re-evaluation of your initial rating. We appreciate any further feedback you may have.

---

### Meta-Review · Area_Chair_39dx · 2023-12-06

**Metareview:**

This is a submission with large disparity in rating (1,6,6,8), where the reviewer who gave rating 1 criticizes the paper mainly on its clarity of representation. Due to the boarderline rating, I've read the paper myself.

This paper proposed to learn a stochastic subgoal representation in HRL instead of a deterministic one, with the claimed motivation being that a deterministic one "hinders the stability and efficiency of hierarchical policy learning". However, why and to what degree deterministic subgoals hinders stability and exploration in HRL is not investigated. There are many unjustified claims like this throughout the paper, where the authors try to motivate each of their design choices, leaving critical readers like myself in doubt about the credibility of other claims made in the paper.

This paper is purely empirical, but experiments are conducted only on small environments, particularly variants of the MuJuCo Ant environment, which as far as I know is a very easier environment where even standard non-hierarchical RL methods work well.

In summary I believe this paper should not be accepted, mainly due to the unprofessional and unjustified claims made throughout the paper, and weak experimental evaluation.

**Justification For Why Not Higher Score:**

NA

**Justification For Why Not Lower Score:**

NA

---

### Decision · Program_Chairs · 2024-01-16

Reject